# Infilling of Missing Rainfall Radar Data with a Memory-Assisted Deep Learning Approach

Johannes Meuer[1], Laurens M. Bouwer[2, 3], Frank Kaspar[4], Roman Lehmann[5], Wolfgang Karl[5], Thomas Ludwig[1], and Christopher Kadow[1]

[1]German Climate Computing Center (DKRZ), Hamburg, Germany
[2]Climate Service Center Germany (GERICS), Helmholtz-Zentrum Hereon, Hamburg, Germany
[3]Institute of Geography, University of Hamburg, Hamburg, Germany
[4]Deutscher Wetterdienst (DWD), Offenbach, Germany
[5]Karlsruhe Institute of Technology (KIT), Karlsruhe, Germany

**Correspondence:** Johannes Meuer (meuer@dkrz.de)

**Abstract.** Incomplete spatio-temporal meteorological observations can result in misinterpretations of the current climate state, uncertainties in early warning systems, or inaccuracies in nowcasting models and can thereby pose significant challenges in hydrology research or similar applications. Traditional statistical methods for infilling missing precipitation data demand substantial computational resources and fail over large areas with sparse data - like temporary outages of weather radars. Although recent machine learning advancements have shown promise in addressing missing meteorological or satellite observations, they typically focus on spatial aspects, overlooking the complex spatio-temporal variability characteristic of precipitation, especially during extreme events. We propose a deep convolutional neural network enhanced with a memory component to better account for temporal changes in precipitation fields. This approach can analyse arbitrary sequences from before and/or after the incomplete observation of interest. Our model is trained and evaluated on the hourly RADKLIM dataset, which features 1-km resolution precipitation data derived from combined radar and weather stations across Germany. By infilling both artifical and actual data gaps of RADKLIM, we demonstrate the model's effectiveness, providing detailed insights into its capabilities during significant rainfall events, such as those in May 2012 and July 2021, including those responsible for the Ahrtal flood. This novel approach represents a step forward in hydrological applications, potentially improving the way we predict and manage water-related events by increasing the accuracy and reliability of precipitation data analysis.

## 1 Introduction

The process of producing accurate climate information is crucial for informing policy as well as for applications in various sectors, e.g. water management or agriculture. For example, nowcasting of events such as thunderstorms, heavy rainfall and snowfall plays a vital role in assessing and planning the management of water resources, flood hazards, urban runoff and climate variability for long-term trends (James et al., 2018; Lang, 2002; Wapler et al., 2012; Teegavarapu et al., 2018). Precipitation data collected by weather radars are an important source of information for such applications, but the reliability and accuracy of these applications depend heavily on the quality of the data. However, temporally and spatially continuous systems are

often plagued by outages that lead to missing values (Teegavarapu et al., 2018; Geiss and Hardin, 2021; Kadow et al., 2020; Barrios et al., 2018) and radars in particular are prone to technical challenges such as blocking of radar beams and near-ground dead zones (Winterrath et al., 2017; Geiss and Hardin, 2021). To illustrate this, consider figure 1, which shows three data

samples taken from the RADKLIM dataset of Germany's meteorological service DWD (Winterrath et al., 2018). This dataset is generated using a process that merges radar observations with weather station measurements to produce continuous spatial and temporal precipitation pattern data across Germany at different temporal resolutions (5min and 1hr). However, due to outages, the radars were not able to monitor precipitation in the shaded grey areas, as shown in the figure.

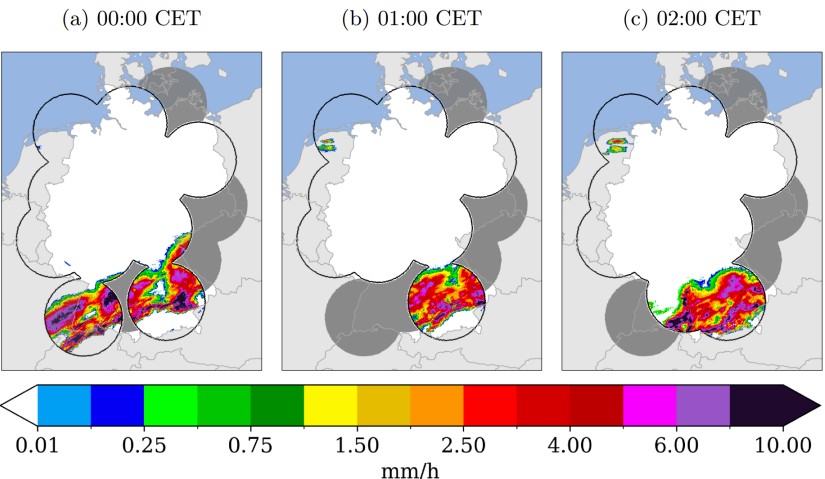

**Figure 1.** Examples of missing precipitation recordings after radar outages in the RADKLIM dataset (Winterrath et al., 2018) on June 24th 2002 of three sequential hours. The grey regions highlight the radars that failed to record any precipitation.

Such problems can lead to misinterpretation or increased uncertainty in observations and predictions. Methods to fill gaps

in climate data range from statistical approaches such as spatial interpolation (Smith et al., 1996; Oliver and Webster, 1990; Teegavarapu et al., 2018; Simanton and Osborn, 1980; Teegavarapu and Chandramouli, 2005; Verworn and Haberlandt, 2011) and linear regression (Vislocky and Fritsch, 1995) to machine learning approaches such as support vector machines (Landot et al., 2008). *Kriging* (Oliver and Webster, 1990), *Inverse Distance Weighting* (Simanton and Osborn, 1980) and *Linear Weight Optimisation Method* (Teegavarapu et al., 2018) provide good estimates for point-wise reconstructions, but their ability to

provide good spatial reconstructions is limited by nearby existing values. This is also shown by Verworn and Haberlandt (2011), who estimate spatial precipitation patterns in northern Germany based on nearby weather stations and radar images. Using only weather stations and statistical reconstruction methods results in overly smooth precipitation maps, while including additional radar information results in much more realistic spatial patterns. Their study highlights not only the limitations of statistical approaches to spatial reconstruction, but also the importance of spatially complete radar images.

Data-driven image inpainting is used to repair image damage caused by raindrops, to improve the quality of old images, or to increase the resolution of low-quality images (Yu et al., 2018; Liu et al., 2018; Elharrouss et al., 2020), but also to fill

gaps in climate data, by training a model on large data-sets to learn the complex patterns within the data. For example, Shibata et al. (2018) use inpainting to reconstruct incomplete satellite images of sea surface temperatures, and Geiss and Hardin (2021) propose a generative adversarial network (GAN) to fill gaps in patchy radar images. Kadow et al. (2020) apply partial convolutions (Liu et al., 2018) to reconstruct missing values in global surface temperature grids. Partial convolutions are able to reconstruct missing values in arbitrarily shaped regions, which is particularly useful when dealing with missing observational data.

While data-driven techniques have shown promising results, they only take into account the spatial relationships in the data, not the temporal variability. On the other hand, while geostatistical approaches take into account data distributions from the past, they lack the ability to create realistic spatial features, especially when considering large regions with missing values or "moving" weather systems that drive rainfall patterns. To address these limitations, we propose a data-driven image inpainting model that is capable of reconstructing arbitrarily shaped missing value regions using partial convolutions by (Liu et al., 2018), while also taking into account the spatio-temporal relationships in the data using a convolutional Long-Short Term Memory (LSTM) approach by (Xingjian et al., 2015).

The remainder of this paper is organized as follows: Section 2 describes the datasets used and the methodological framework implemented for the infilling of missing precipitation data. Section 3 presents the experimental results and provides a comprehensive discussion of the performance of the proposed models. Finally, Section 4 summarizes the key findings of the study and discusses potential future research directions. The appendices provide additional details on the models' architectures, evaluation metrics, experimental overview, and additional results.

## 2 Data and methods

### 2.1 Data

This study is based on the RADKLIM-dataset of Deutscher Wetterdienst (DWD; Germany's national meteorological service). RADKLIM is a data set of reprocessed gauge-adjusted radar data (Winterrath et al., 2017, 2018). DWD operates a network of 17 weather radar stations (C-band; Lengfeld et al. (2019)) as well as a network of several hundred ground-based rain gauges (Kaspar and Mächel, 2023). Weather radars send pulses of electromagnetic energy into the atmosphere and measure reflectivity to monitor the movement, intensity and type of precipitation, such as rain, snow and hail. However, radars cannot observe precipitation directly. In contrast, rain gauges can only provide incomplete spatial information of precipitation events, depending on the density of the network. To derive quantitative precipitation estimates (QPEs), the signals from weather radars can be combined with measurements from the rain gauges. To provide timely information, esp. for water management applications, DWD derives QPEs in real-time by adjusting radar observations with data from the German rain gauge network (RADOLAN: Radar Online Adjustment). The information is stored as precipitation intensities with an intensity resolution of 0.1 mm and a spatial resolution of 1 km$^2$ resulting in a 900x1100 km grid for Germany, and is provided in real time within 30 minutes of the last measurement. The data are provided at five-minute and hourly time frequencies. The products might contain gaps caused by outages of radars, e.g. due to technical failures or regular maintenance. The archived collection

of operationally produced RADOLAN data is also inhomogeneous in time as it was produced with the current hardware and software configuration at the time of creation. To provide a dataset suitable for climatological application, the radar reflectivities have been reprocessed (i.e., the same set of algorithms for artefacts and attenuation correction as well as adjustments to rain gauge observations has been applied) to create a homogeneous set of precipitation observations covering the period from 2001 onwards ("RADKLIM", Lengfeld et al. (2019)). In this study we focus on the hourly grids of RADKLIM (Winterrath et al.,
2018) (based on RADOLAN RW) for training and evaluation spanning 2001-2022.

## 2.2 Model

### 2.2.1 Image Inpainting U-Net

We use an image inpainting U-net (Ronneberger et al., 2015) as a baseline model for reconstructing missing value regions in precipitation data. This model takes a single precipitation grid as input. Each data sample contains an observed precipitation
grid with no missing values (i.e. the original RADKLIM data, considered as 'ground truth' in our analysis), a binary missing value mask, and a masked grid obtained by element-wise multiplication of the original observations and the mask. This allows us to simulate missing value regions in complete radar images. We replace conventional convolutional operations in the U-net with partial convolutions by Liu et al. (2018). The automatic mask updating mechanism in these operations efficiently handles irregularly shaped holes in missing value masks, outperforming other methods trained only on regularly shaped holes (Liu
et al., 2018). Equation 1 defines partial convolution operations that show improved inpainting results, especially for large and irregularly shaped missing value regions. The first term describes the masking of the input data $I$ with the mask $M$ and applies a scaling factor depending on the number of ones in $M$. The weight matrix $W$ represents the trainable weights of the network. The second term describes the mechanism for updating the mask after each convolution. If the convolution has been able to apply its output to at least one valid input value, that position is marked as valid. The architecture of our baseline convolutional
neural network (CNN) is illustrated in Appendix A1.

$$
\begin{aligned}
i' &= \begin{cases} W^T(I \circ M)\frac{sum(1)}{sum(M)} + b, & \text{if } sum(M) > 0 \\ 0, & \text{otherwise} \end{cases} \\
m' &= \begin{cases} 1, & \text{if } sum(M) > 0 \\ 0, & \text{otherwise} \end{cases}
\end{aligned} \tag{1}
$$

### 2.2.2 Temporal Memory Module

Precipitation estimation is a challenging spatio-temporal problem, as it involves highly non-linear patterns in both time and
100 space (Xingjian et al., 2015; Tian et al., 2019). To incorporate temporal information, we propose a straightforward channel-based approach that includes sequences of precipitation grids within the input data, rather than considering only grids of single timesteps. We can arbitrarily define the number of time steps to consider. The dimension of the channel is equal to the number

of time steps considered as input data. The output of the network is a tensor representing a reconstructed precipitation grid for a single time step. The detailed architecture can be seen in Appendix A2.

This channel-based approach has limitations in distinguishing between long-term and short-term relationships in the data. To address this, Xingjian et al. (2015) proposed a convolutional long short-term memory (LSTM) network for precipitation nowcasting, which extends the original fully connected LSTM architecture of Hochreiter and Schmidhuber (1997) with convolutional structures. The network takes as input a sequence of time-continuous precipitation grids collected from weather radars. The equations of the fully connected LSTM are modified to incorporate convolutional operations, given by Equations 2.

$$f_t = \sigma(W_{fx} * X_t + W_{fh} * H_{t-1} + W_{fc} \circ C_{t-1} + b_f)$$
$$i_t = \sigma(W_{ix} * X_t + W_{ih} * H_{t-1} + W_{ic} \circ C_{t-1} + b_i)$$
$$g_t = tanh(W_{gx} * Xt + W_{gh} * H_{t-1} + b_g)$$
$$C_t = f_t \circ C_{t-1} + i_t \circ g_t$$
$$o_t = \sigma(W_{ox} * X_t + W_{oh} H_{t-1} + W_{oc} \circ + b_o)$$
$$h_t = o_t \circ tanh(C_t) \tag{2}$$

    The $W$s and $b$s are weights and biases that are learned by the model. The memory states $C_t$, $C_{t-1}$, the hidden states $H_t$, $H_{t-1}$ and the gates $f_t$, $i_t$, $o_t$, $g_t$ are represented as three-dimensional matrices, with input channels in the first dimension and a two-dimensional spatial precipitation field in the other two dimensions. The Hadamard product is denoted by $\circ$ and the convolutional operation is denoted by $*$. The convolutional LSTM is implemented in an encoder-decoder architecture similar to the future predictor model proposed by Srivastava et al. (2015). The combination of spatial data processing using CNNs and sequential data processing using LSTMs has been shown to outperform each technique used individually.

    Instead of processing the data sequences in the channel dimension, the input sequence is iteratively passed through the set of equations 2. Note that these operations are applied only to the precipitation tensors, not to the mask tensors. Similar to the autoencoder of Srivastava et al. (2015), the decoder reconstructs the input sequence in reverse order. As a result, the output tensors from the encoder are provided to the decoder in reverse order. This speeds up the weight optimisation, as the model primarily considers short-range correlations Srivastava et al. (2015). Appendix A3 shows how we implemented the LSTM module.

### 2.3   Technical Implementation

The models presented in this study were implemented in Python using the PyTorch deep learning framework by Facebook, chosen for its flexibility and efficiency in developing and training neural networks. PyTorch's dynamic computation graph and extensive support for GPU acceleration made it an ideal choice for training the models.

    For model training and evaluation, we utilized A100 80 GB GPUs, which provided the necessary computational power for handling large datasets and complex model architectures. The use of these high-performance GPUs enabled efficient processing

and reduced training time for both the baseline and more computationally intensive LSTM models. The models were trained on high-performance hardware to ensure that they could effectively scale to meet the demands of large-scale image inpainting tasks and complex climate data gap-filling.

## 2.4 Experimental Design

Due to radar outages, which are still quite common, many grids from the RADKLIM dataset contain missing values. We have therefore created a dataset containing only complete radar images where no radar outages have occurred, and a dataset containing missing values where radar outages have occurred. In addition, the DWD introduced three new radar stations in 2014/2015 to the original 14 radar stations to improve spatial coverage. Therefore, we performed separate analyses on data from only 14 radars over the complete time range and from all 17 radars from 2015 onwards.

An overview of the different experiments can be found in Appendix C1, where we trained three models on each experiment: the baseline, channel-based and LSTM architectures.

In order to comprehensively evaluate our proposed models, our first experiment consists of a single missing value mask covering a significant area in central Germany for the first experiment. This gap was caused by three radar outages in January 2012, resulting in a three-hour data absence (see Appendix G). The chosen area presents an additional challenge as it encompasses the Harz Mountains, renowned for intricate precipitation patterns influenced by terrain (Panziera et al., 2011).

In the second experiment, we trained the three models on all the complete hourly data from 2001-2022 covered by 14 radars. We extracted a mask dataset by setting all missing values from the remaining incomplete hourly data to zero and the existing values to one. During training, we combined the complete data with all the extracted missing value masks. This gave us the most reliable models for infilling real case scenarios with missing values.

In our third and final experiment, we looked at a recent flood event caused by extreme precipitation, training our models on complete data from 2015-2018, covered by all 17 radars. In July 2021, Germany was hit by massive rainfall events and the resulting floods caused more than 180 deaths and billions of euros in damage. The return period of the 24h rainfall is estimated to be around a 500 years, with an even longer return period in the most affected area (Mohr et al., 2023).

## 3 Results and discussion

Our performance evaluation relies on a range of metrics: (1) root mean square error (RMSE in mm/h), (2) absolute mean error (AME in mm/h), (3) temporal correlation over spatial mean ($r_{xy,time}$), (4) mean spatial correlation ($\overline{r_{xy,space}}$), and (5) spatial correlation of sum over time ($r_{xy,sum}$). Detailed explanations of these calculations are provided in Appendix B. The models were trained on complete grids from 2001 to 2022 covered by 14 radars and excluding the evaluation years, utilizing the missing data mask from Appendix G. We specifically chose the years 2007, 2012, and 2016, which were not part of the training data, to allow meaningful comparisons across three distinct annual cycles. Metrics were then computed by infilling simulated missing values for 2007, 2012, and 2016 with the same mask and averaging the results across these years.

Figure 2 (detailed in Table D1 shows that the channel-based model outperforms the baseline model in terms of pixel-wise (RMSE) and average (AME) precipitation reconstruction and maintains a good temporal correlation. However, the baseline model outperforms the channel-based model in terms of spatial metrics ($\overline{r_{xy,space}}$ and $r_{xy,sum}$), indicating that it is better at reconstructing the spatial distribution per time step as well as the total amount of precipitation at a fixed location. This may be due to the under-estimation of precipitation in the channel-based approach, which can be seen in figure 3. The final model

including the temporal memory module is the best performing model with the highest scores for temporal, spatial and summed correlations, and lowest errors (RMSE and AME).

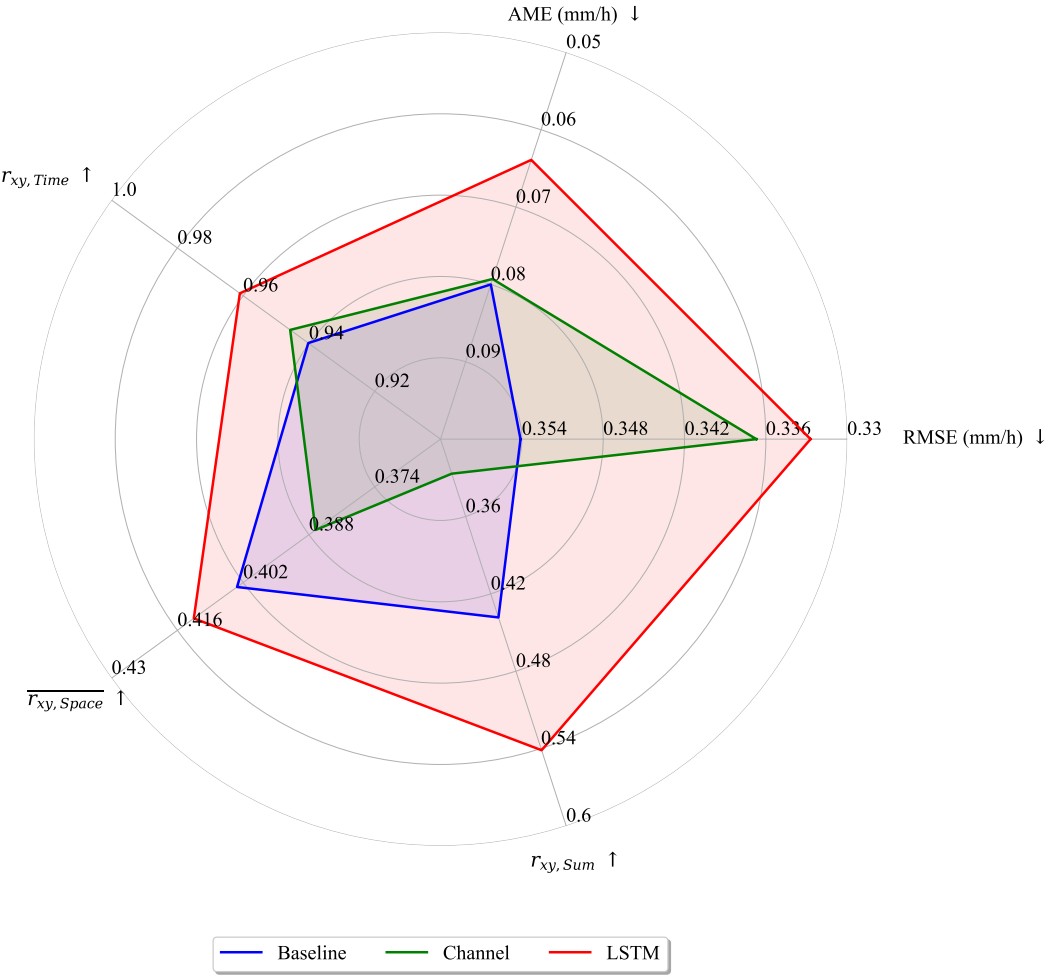

**Figure 2.** Verification metrics of the baseline model (blue) with the channel-based memory approach (green) and the advanced temporal memory module (red). The ↑ determines that a high - and the ↓ a low value should be aimed for.

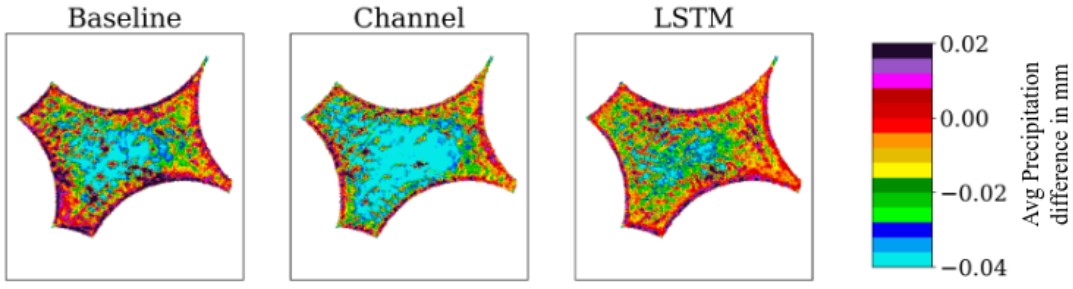

**Figure 3.** Average difference between the predicted precipitation in the infilled area and the actual precipitation measured. The field is an average of all infilled grids from 2007, 2012 and 2016. A negative value means that too little precipitation was predicted and a positive value means that too much was predicted.

We further explore the results of the first experiment using scatter plots in Figure 4, which provides a visual comparison between the model predictions and the observations. The baseline approach exhibits the largest spread with respect to the spatial average precipitation, while the channel approach underestimates especially average fields with low precipitation. Here, the LSTM model provides the most accurate results. In the matter of reconstructing the temporal average at each grid point, the results of the baseline model feature the worst overall approximation and major outliers between 0.05 and 0.1 mm/h, indicating over-estimation of precipitation at specific grid points. The channel-based approach reduces the number of outliers in this range but still exhibits overestimations between 0.1 and 0.15 mm/h, as well as underestimations at most grid points. The LSTM provides the best estimations for the in time averaged grid points being closest to the red line with much smaller outliers.

In Figure 5, we make a visual comparison of reconstructed grids for this single missing value mask scenario. Here, we consider a rainfall event from May 2012. The results from the channel-based approach again show an underestimation of precipitation during the event. On the other hand, the baseline and LSTM approaches give very similar results compared to the original RADKLIM data. However, the LSTM has some artificial checkerboard patterns, which were also observed by Liu et al. (2018) from the original partial convolutions.

Figure 6 shows results from our second experiment, a visual comparison of infilled images generated by the baseline and LSTM models from the June 2002 radar outages, where we have no original observations. The baseline model was provided with a single time step for each infill application, while the LSTM was additionally provided with the two previous and subsequent time steps. Comparing images (a) and (b) from the baseline model (centre), a large region of precipitation disappears within one hour. In contrast, the LSTM model is able to preserve this by incorporating the temporal information of the previous two hours. It is clear that the LSTM model produces the most realistic results, while the spatial patterns are also best approximated with this model.

Figure 7 shows the results from the third experiment, which is a snapshot of the Ahrtal rainfall event (top row). We masked out a large region of precipitation west of the location of the largest flooding (Ahr river basin) and applied the LSTM model

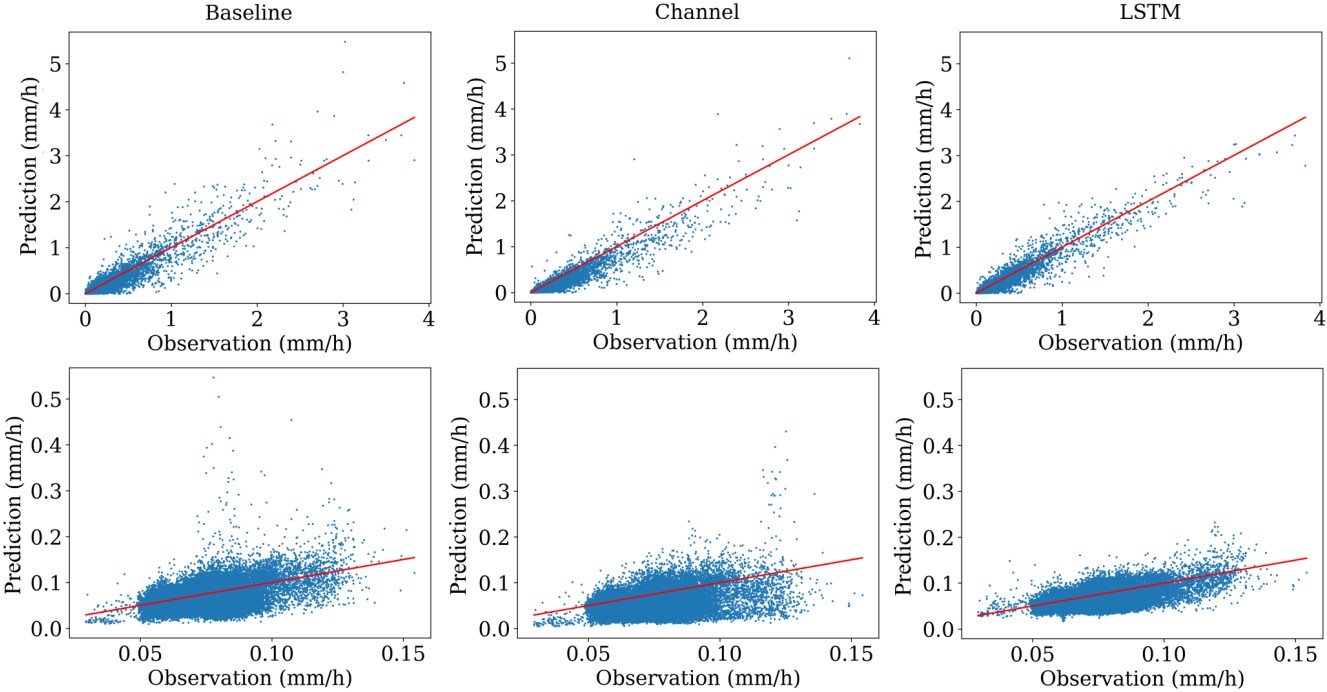

**Figure 4.** The top row shows the spatial average of precipitation for each time step from 2007, 2012 and 2016. The bottom row shows the temporal average of all grids in 2007, 2012 and 2016 for each grid point. Both calculations were performed on the infilled area only. The red line marks the optimal results.

to infill this region. We have also applied the infilling of this region to 20 consecutive hours of the extreme rainfall event and show the accumulated rainfall over this period (bottom row). A complete map of the event and the reconstruction can be seen in Appendix E. Here we can see a limitation of our model, which leads to an underestimation of the extreme rainfall. The model predicts a total rainfall of 70.76 mm in the Ahr basin over the 20 hour period, whereas the original observed rainfall was 101.89 mm. This amounts to an accuracy of 69.5 %. However, in the event of radar failure during such events, our method could still potentially provide an estimate of rainfall over the affected region based on nearby radar data and help improve flood forecasting and risk assessment.

The evaluation of the proposed models shows that the LSTM model outperforms the baseline and channel-based approaches in terms of overall accuracy. The model consistently improves performance across all evaluation metrics (figure 2), as well as through visual analysis of scatter plots in figure 4 and infilled images in figure 6. However, it should be noted that the LSTM model also requires significantly more hardware resources and computing time for training compared to the other models. For example, the baseline model took approximately 10 hours to train for 100k iterations on an A100 GPU, consuming around 10 GB of GPU memory. In contrast, the LSTM model required roughly 40 hours for the same number of iterations, consuming 60 GB of GPU memory. During inference, this time is greatly reduced, with the baseline model taking about 1 second per

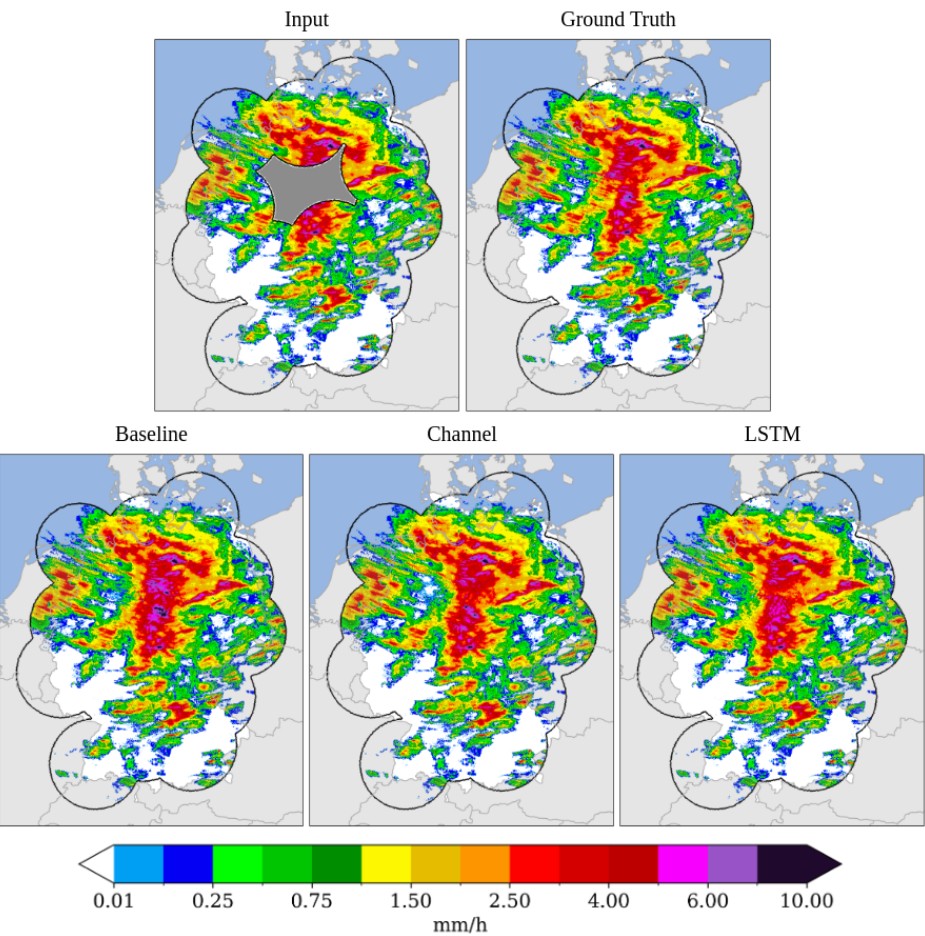

**Figure 5.** The top row shows the input to the models (simulated missing value region in grey) and the original RADKLIM data. The bottom row shows the infilled grids from our three models. This exact event was recorded on 31st May 2012 18:00 CET.

prediction and the LSTM model taking 5 seconds per prediction. Nevertheless, this may impact the feasibility and scalability
of implementing the LSTM model in certain operational settings with limited computing resources. Hence, a careful trade-off
analysis may be necessary when choosing the most appropriate model for a specific application.

In the final step, we filled in all hourly data from 2001 to 2022 using our LSTM model, which was trained on data from
17 radar stations. This enhanced dataset extends radar coverage to periods before 2014, ensuring both spatial and temporal
consistency. The infilled dataset will be made available for public access upon publication of this study. A few selected samples
from this dataset are shown in Appendix G1.

The results of our study demonstrate that the memory-assisted deep learning approach, particularly the LSTM-based model,
outperforms the baseline convolutional models in infilling missing radar rainfall data. This indicates that incorporating tempo-

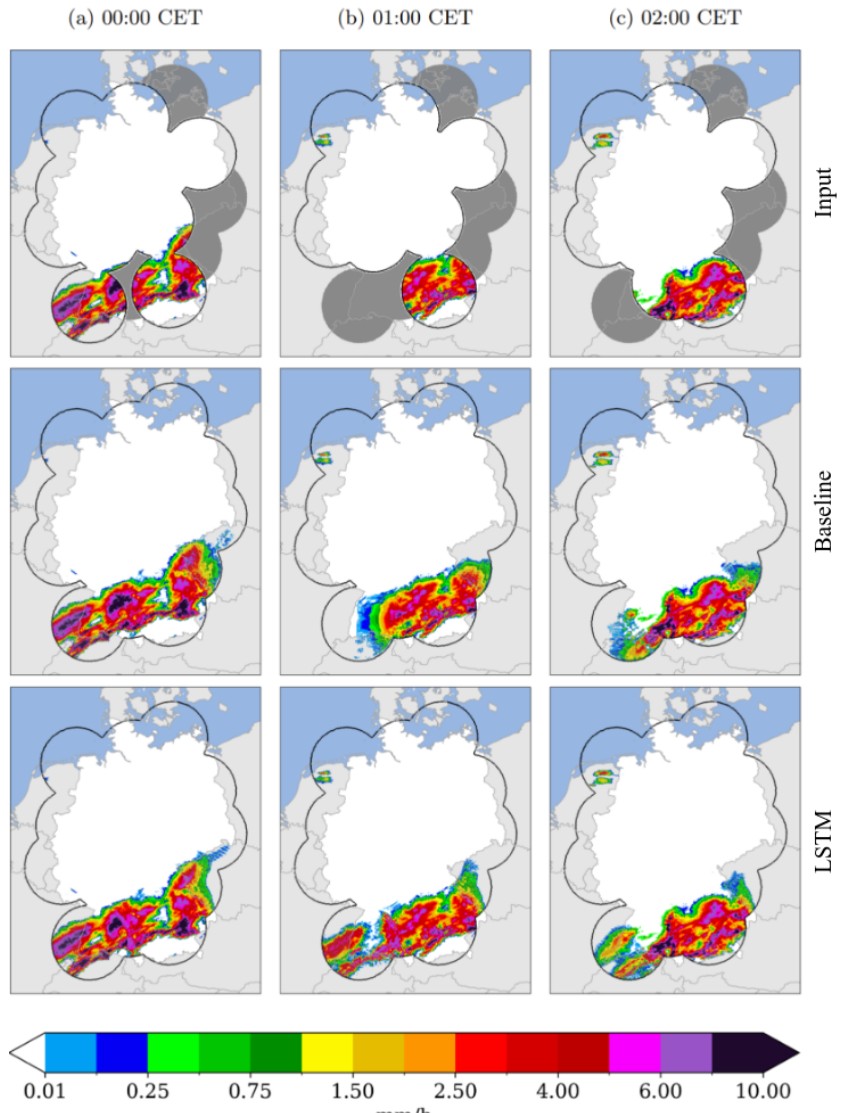

**Figure 6.** Comparison of the patchy rainfall records (top row) used as input to our infilling models with the reconstructed results from the baseline model (centre row) and the LSTM model (bottom row).

ral dependencies significantly enhances reconstruction accuracy. This highlights the importance of capturing temporal dependencies in precipitation reconstruction. Furthermore, our deep learning approach avoids the excessive smoothing observed in

traditional interpolation methods, leading to more realistic precipitation patterns.

Despite the good performance, our approach has certain limitations. One notable issue is the underestimation of extreme rainfall events, as observed in the Ahrtal flood case study. This may be due to the known inherent difficulty of machine learning models in predicting rare and highly localized extreme values, which are often underrepresented in the training data

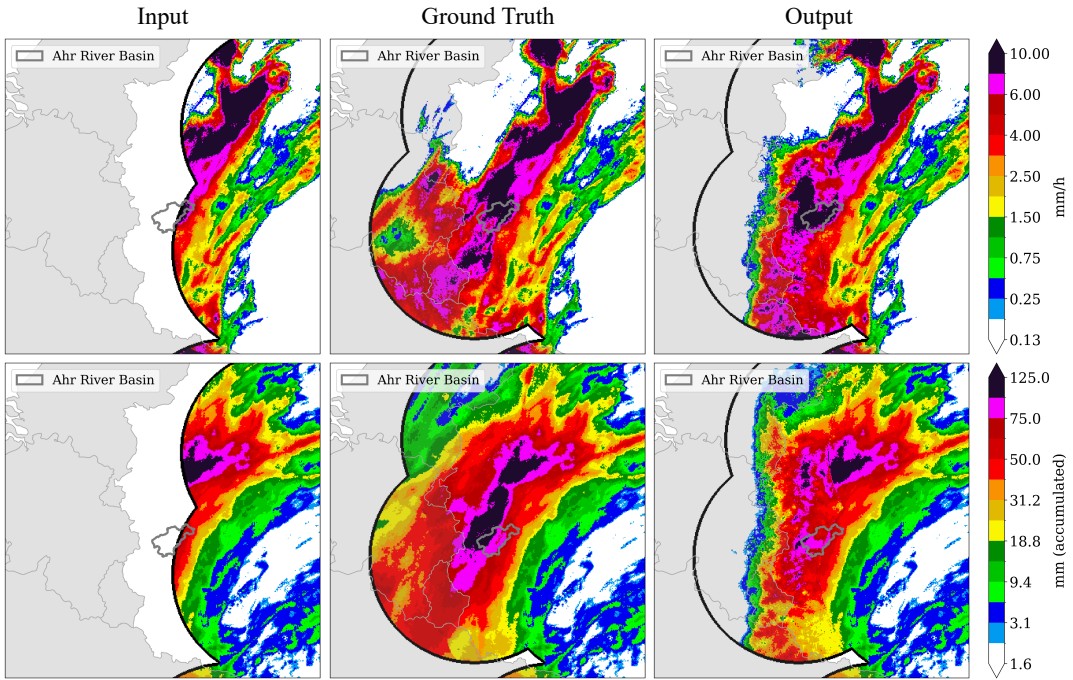

**Figure 7.** A zoomed in illustration of the extreme precipitation event in the Ahrtal on the 14th July in 2021. The entire area can be seen in figure E. The Ahr river basin is highlighted by gray border. Here we compare the original RADKLIM data with the output of the LSTM model. The top maps show a reconstruction of a single hour (18:00 CET) in mm/h, the bottom row shows the accumulated amount of precipitation in mm from reconstructions over 20 hours (between 2:00 and 21:00 hrs. on 14 July, 2021).

(Charlton-Perez et al., 2024; Xu et al., 2024; Zhong et al., 2024). Future work could address this by incorporating additional
features such as atmospheric pressure, wind speed, temperature, and humidity fields.

Another limitation is the computational cost of training and deploying LSTM-based models. While they outperform simpler methods, their training process requires significant computational resources, which may not be feasible for operational nowcasting systems with real-time constraints. A potential trade-off could involve using a hybrid approach where simpler models handle routine infilling, while the LSTM model is reserved for high-impact scenarios.

Nevertheless, the implications of these findings are significant for hydrology and meteorology. Accurate rainfall reconstructions are crucial for flood forecasting, climate monitoring, and weather prediction models. By improving the reliability of precipitation data, our method can enhance the accuracy of early warning systems and support better decision-making for disaster response and water resource management:

    1. More accurate rainfall estimates can lead to better flood predictions, reducing uncertainty in early warning systems.

2. Long-term datasets with fewer gaps allow for better trend analysis and climate change assessments.

3. Reliable precipitation data informs irrigation planning and drought mitigation strategies.

Our method can also be integrated into existing operational meteorological systems, helping to mitigate the impact of radar outages and ensuring continuity in rainfall monitoring.

While our results are promising, several avenues for future research could enhance the model's performance and applicability:

1. Hybrid Models: Combining physics-based numerical weather prediction models with deep learning techniques could improve performance and efficiency, particularly for extreme events.

2. Transfer Learning: Training the model on a diverse set of meteorological data, including satellite and ground-based observations, could enhance its generalizability to different regions and climate conditions.

3. Attention-Based Models: Exploring attention mechanisms, such as those used in transformer architectures, could help capture long-range dependencies more effectively and efficiently than LSTMs.

4. Multi-Source Data Integration: Incorporating additional meteorological variables (e.g., temperature, wind speed, atmospheric pressure) may improve model accuracy by providing a more comprehensive representation of precipitation dynamics.

5. Operational Deployment: Research into optimizing model efficiency for real-time applications could make deep learning-based infilling viable for nowcasting systems.

## 4 Conclusions

Incomplete precipitation data can lead to misinterpretations of climate conditions, uncertainties in early warning systems, and inaccuracies in nowcasting models, posing challenges for hydrological and meteorological applications. Addressing this issue, we proposed and evaluated three machine learning models for infilling missing precipitation data: A basic inpainting model, a straightforward channel-based approach that considers a sequence of time steps as input, and an LSTM approach as an advancement to the channel-based approach. We trained and evaluated our models on hourly precipitation data over Germany. The results of the evaluation showed that the LSTM model outperforms the baseline and channel-based approaches in terms of overall performance, the baseline model has a larger prediction error, and the channel-based model has a tendency to underestimate precipitation. However, the LSTM model also requires significantly more hardware resources and computing time compared to the other models, while the other models already perform quite well. Increasing the complexity of the model can improve the results, with the understanding that it requires more resources than other models. Our analysis highlights the potential of machine learning models to be used for efficiently infilling missing precipitation data. The tangible advantage of employing the infilled data generated by our models requires exploration through an examination of how it might enhance the accuracy and performance of other nowcasting models that depend on the same dataset. Future applications of our research

could include a cascaded approach, using the baseline model for immediate results, the channel-based model for better temporal estimation, and finally the best results with the LSTM model with the largest latency. We also see scope for additional applications, beyond the infilling of missing radar data. By combining information on observed rainfall data from rain gauges, additional spatial information could be gained and the accuracy of the rainfall information can be improved.

*Code and data availability.* The code used for the analysis and simulations in this study is openly accessible at: https://github.com/FREVA-CLINT/ climatereconstructionAI. The repository contains scripts written in Python, along with detailed documentation to facilitate replication and further exploration of the methodologies employed. The data utilized in this research were obtained from the Germany's meteorological service (Deutscher Wetterdienst, DWD). Hourly radar-based precipitation data (RADKLIM) covering the study period are available under an open data license: http://dx.doi.org/10.5676/DWD/RADKLIM_RW_V2017.002. A fully infilled version next to the original data of the

hourly precipitation data will be made available to the public upon publication of this study.

## Appendix A: Model

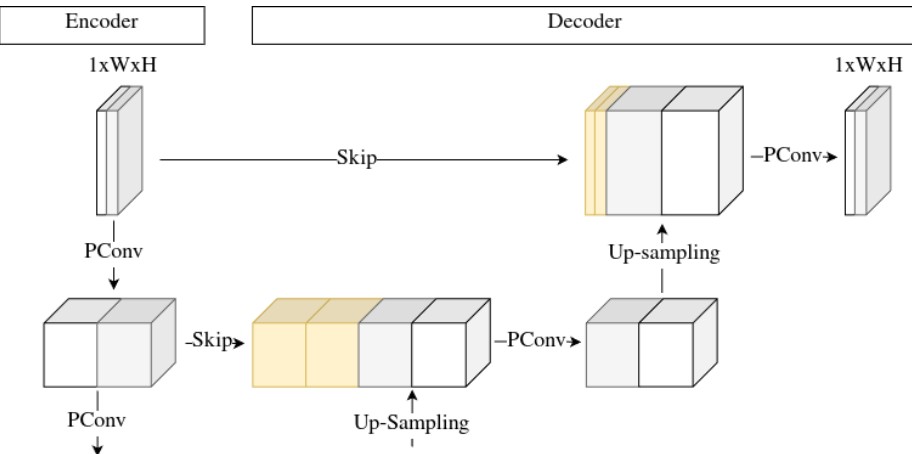

**Figure A1.** The architecture of the baseline model.

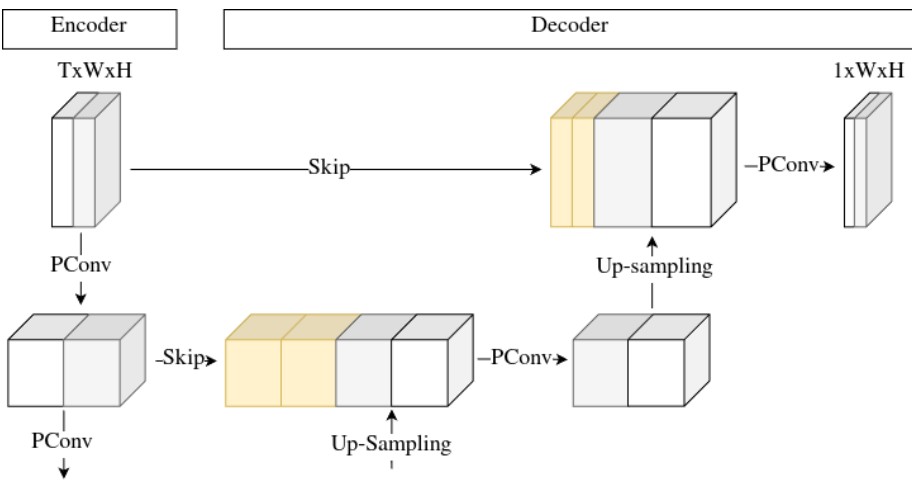

**Figure A2.** The architecture of the channel-based model.

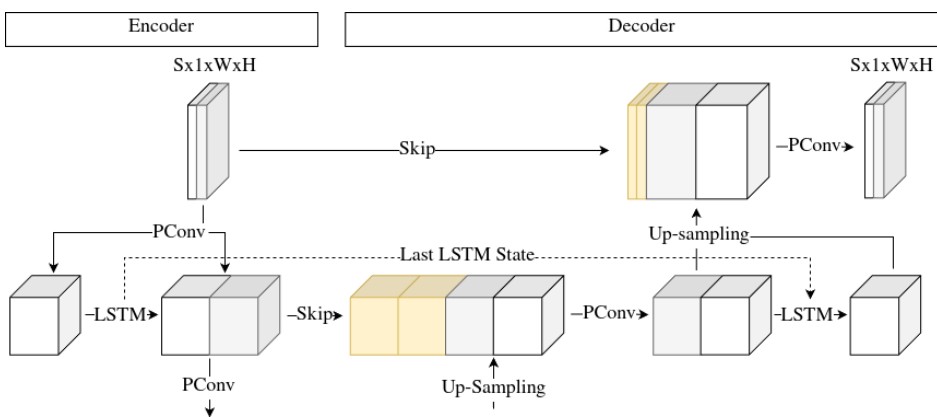

**Figure A3.** The architecture of the LSTM model.

## Appendix B: Formulas

$$RMSE = \sqrt{\frac{1}{N \cdot W \cdot H} \sum_{t=0}^{N-1} \sum_{i=0}^{W-1} \sum_{j=0}^{H-1} (x_{t,i,j} - y_{t,i,j})^2} \tag{B1}$$

$$AME = |\frac{1}{N} \cdot \sum_{t=0}^{N-1} (\frac{1}{W \cdot H} \sum_{i=0}^{W-1} \sum_{j=0}^{H-1} x_{t,i,j} - \frac{1}{W \cdot H} \sum_{i=0}^{W-1} \sum_{j=0}^{H-1} y_{t,i,j})| \tag{B2}$$

$$r_{xy,Time} = \frac{\sum_{t=0}^{N-1} T_{x,t} \cdot T_{y,t} - \overline{T_x} \cdot \overline{T_y}}{\sqrt{(\sum_{t=0}^{N-1} T_{x,t}^2 - \overline{T_x})(\sum_{t=0}^{N-1} T_{y,t}^2 - \overline{T_y})}} \tag{B3}$$

$$r_{xy,Space} = \frac{\sum_{i=0}^{W \cdot H - 1} x_{t,i} \cdot y_{t,i} - \overline{x_t} \cdot \overline{y_t}}{\sqrt{(\sum_{i=0}^{W \cdot H - 1} x_{t,i}^2 - \overline{x_t})(\sum_{i=0}^{W \cdot H - 1} y_{t,i}^2 - \overline{y_t})}} \tag{B4}$$

$$r_{xy,Sum} = \frac{\sum_{i=0}^{W \cdot H - 1} \Sigma(x_i) \cdot \Sigma(y_i) - \overline{\Sigma(y_i)} \cdot \overline{\Sigma(y)}}{\sqrt{(\sum_{i=0}^{W \cdot H - 1} \Sigma(x_i)^2 - \overline{\Sigma(x)})(\sum_{i=0}^{W \cdot H - 1} \Sigma(y_i)^2 - \overline{\Sigma(y)})}} \tag{B5}$$

# Appendix C:  Experiment Overview

**Table C1.** An overview of all experiments that we performed.

| Experiment | Training time range | Evaluation time range | Training samples | Evaluation samples | Radar Coverage |
|---|---|---|---|---|---|
| 1 - Single Mask | 2001-2022[1] | 2007, 2012, 2016 | 55989 | 26328 | 14 |
| 2 - All Masks | 2001-2022 | 2001-2022 | 82317 | 87263 | 14 |
| 3 - Ahrtal | 2015-2018 | 2021 | 15647 | 20 | 17 |

1 Excluding years 2007, 2012 and 2016

## Appendix D: Metric Results

**Table D1.** Verification metrics of the baseline model, the channel-based memory approach, and the advanced LSTM approach. The ↑ in the columns determines that a high - and the ↓ a low value should be aimed for. The bold text highlights the best performing model in the channel based models and the advanced approaches.

| Model | RMSE ↓ | AME ↓ | $r_{xy,Time}$ ↑ | $\overline{r_{xy,Space}}$ ↑ | $r_{xy,Sum}$ ↑ |
|---|---|---|---|---|---|
| Baseline | 0.3541 | 0.08 | 0.9402 | 0.4033 | 0.4383 |
| Channel | 0.3367 | 0.0793 | 0.9457 | 0.3866 | 0.3268 |
| LSTM | **0.3327** | **0.0639** | **0.961** | **0.4125** | **0.5413** |

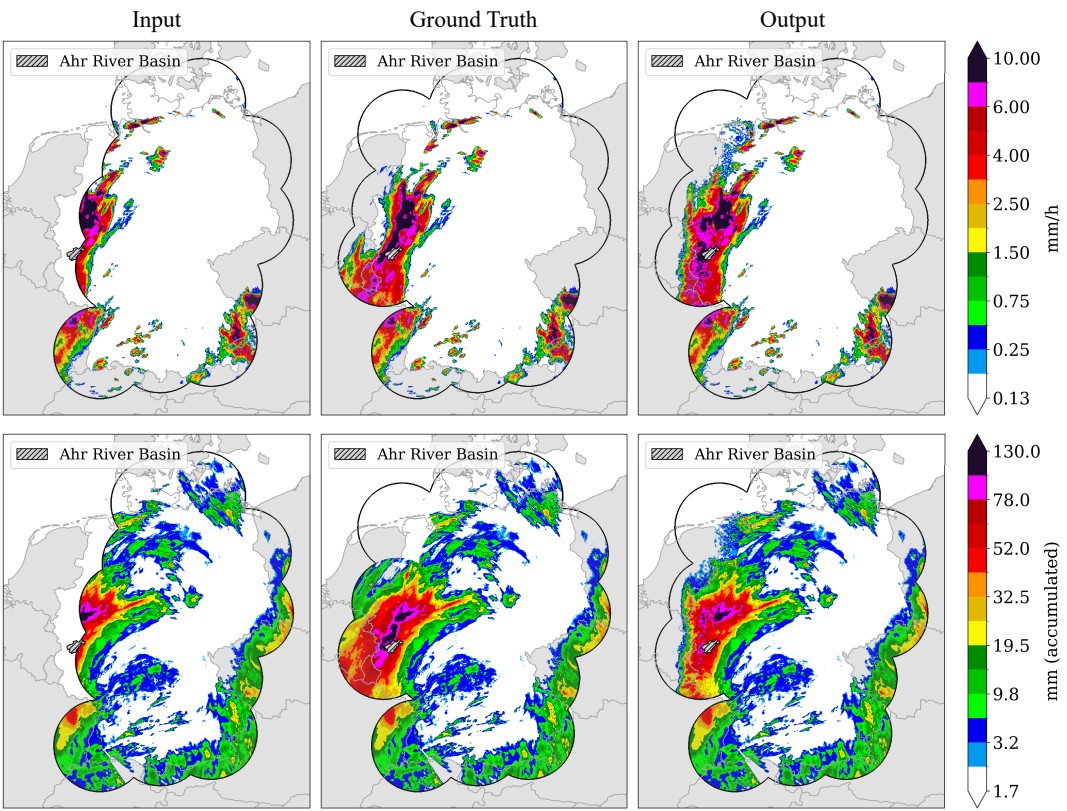

**Figure E1.** Illustration of the extreme precipitation event in the Ahrtal on the 14th July in 2021 over the entire area of Germany. The Ahr river basin is highlighted by gray hatching. Here we compare the original RADKLIM data with the output of the LSTM model. The top maps show a reconstruction of a single hour (18:00 CET) in mm/h, the bottom row shows the accumulated amount of precipitation in mm from reconstructions over 20 hours (between 2:00 and 21:00 hrs. on 14 July, 2021).

**Appendix F:  Training Mask**

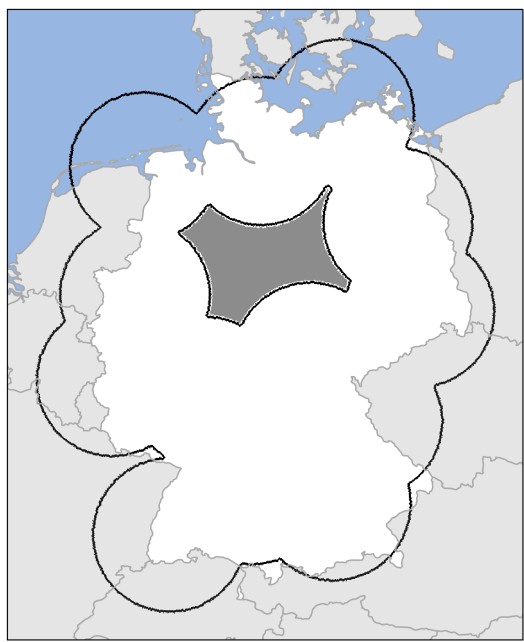

**Figure F1.** Real case study of radar outages that occurred in January 2012. Four overlapping radars failed at the same time, causing a large region with missing values (grey) in the precipitation recording.

## Appendix G: Infilled Samples

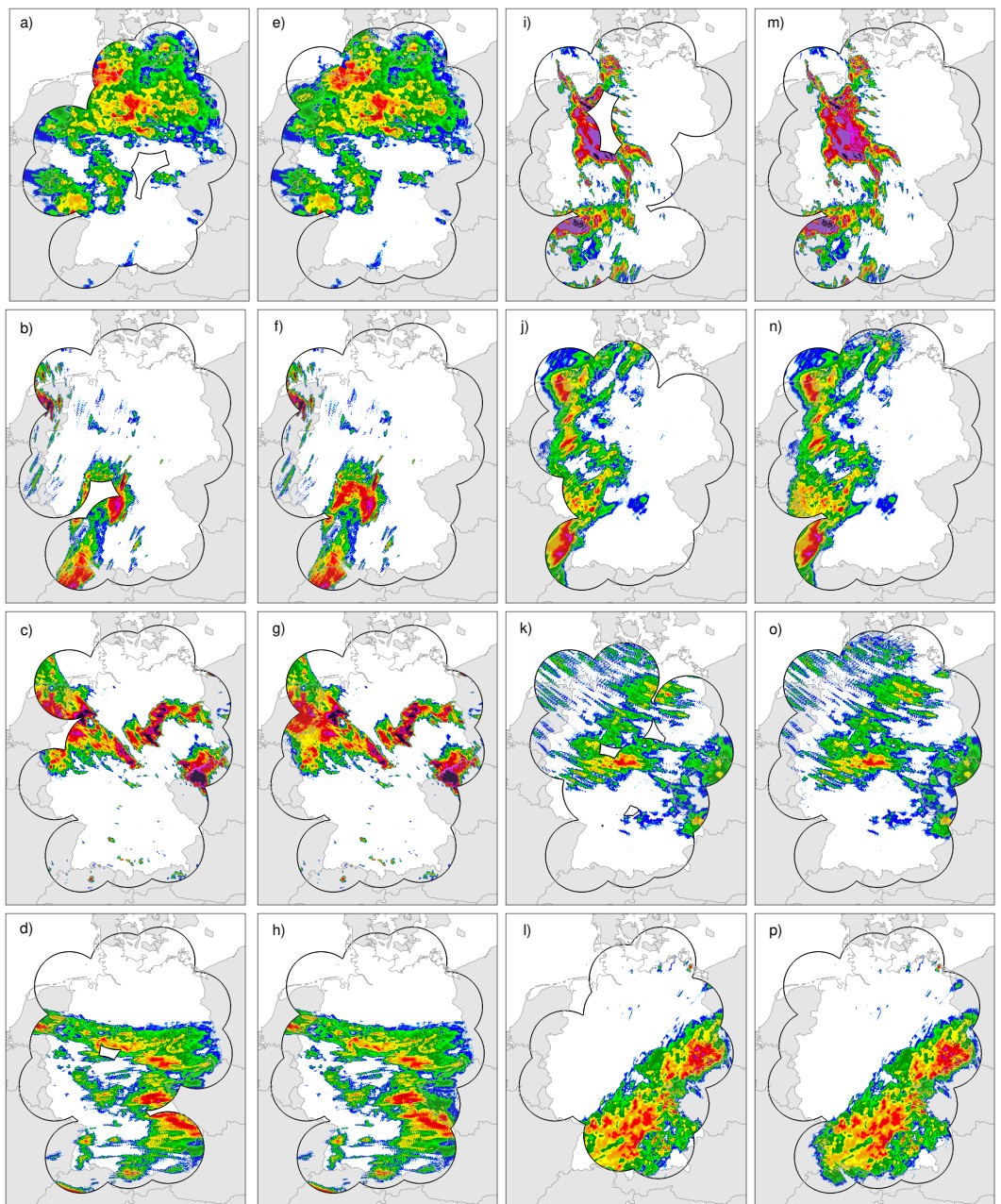

**Figure G1.** A selection of infilled samples using the LSTM model. a)-d) and i)-l) show samples from the original dataset, and e)-h) and m)-p) the corresponding infilled versions.

*Author contributions.* Johannes Meuer, Laurens Bouwer, and Christopher Kadow conceived the study. Johannes Meuer designed the research methodology, conducted data analysis, and collected and processed the data used in the study and wrote the paper. All authors contributed to the interpretation of results and provided critical feedback throughout the manuscript preparation. All authors contributed to the writing and editing of the manuscript.

*Competing interests.* The authors of this article declare that no competing interests are present.

*Acknowledgements.* The authors are grateful to the German Climate Computing Center (DKRZ) for providing the hardware for the calculations and to the Germany's meteorological service (DWD) for providing the RADKLIM data that were used for this study. This work was supported by the Horizon Europe project EXPECT (Towards an Integrated Capability to Explain and Predict Regional Climate Changes) under Grant Agreement 101137656 (C.K.). Johannes Meuer is funded from the German Science Foundation (DFG), provided by the research unit FOR 2820, titled "Revisiting The Volcanic Impact on Atmosphere and Climate-Preparations for the Next Big Volcanic Eruption" (VolImpact), with the project number 398006378. Johannes Meuer, Roman Lehmann and Wolfgang Karl acknowledge support by the Karlsruhe Institute of Technology (KIT). Thomas Ludwig and Christopher Kadow are funded from the German Climate Computing Center (DKRZ). Laurens Bouwer acknowledges support by the Helmholtz Association under the research program "Changing Earth – Sustaining our Future".

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
