# Peer review of "Infilling of Missing Rainfall Radar Data with a Memory-Assisted Deep Learning Approach"

_EGUsphere, 2024_

## Author Response (AR1)

**Response HESS manuscript egusphere-2024-1392**

Anonymous Referee #1

The study of Meuer et al entitled "Infilling of Missing Rainfall Radar Data with a Memory-Assisted Deep Learning Approach" explores the use of machine learning for infilling of the RADOLAN precipitation product of the German Weather Service. The approach comprises a comparison with baseline and channel-based results. Although the proposed method is technically more demanding, it results in increased accuracy and in filled data gaps respecting the temporal and spatial pattern of the precipitation event.

The manuscript is written in a concise way, is well-structured and makes use of good English. However, to my impression, the actual discussion is missing and only few remarks could be interpreted as discussion. Therefore, please enlarge the discussion part and make use of more references to guide the interpretation. In this context, a broader discussion on the uncertainty of each infilling technique would also help to improve this section. Why were the events of May 2012 and July 2021 presented only?

Response: We thank the reviewer for the thorough review and for these kind remarks. We agree to extend the discussion section, and to include a treatment of each of the infilling techniques. In addition, we would like to explain that the May 2012 and July 2021 events were chosen because they represent an "average" rainfall event (2012) and a very extreme rainfall event (2021), and thereby provide a good illustration of the capabilities of the algorithms.

In addition, we provide information on other periods, as shown in Figures 3 and 4, for the years 2007, 2012 and 2016.

To conclude, the manuscript requires further improvements my opinion before recommending it for publication in this journal.

Response: we thank the reviewer for making the points above, and the editorial comment below, which we will include in the revision of our paper.

Specific comments:

p.1, l. 2: "challenges"

Response: Thanks, we corrected this.

p.2, l. 24: Please correct the figure number

Response: We adjusted to "Figure 1".

p.3, l. 64: "depending" instead of "pending"

Response: We corrected this.

p.3, l. 70: Please add the specific product name (RADOLAN YW and RADOLAN RW)

Response: We added a sentence, stating that we are using the RW product.

p.4, l. 90: Please introduce the abbreviation

Response: we added "Convolutional Neural Network (CNN)".

p.4, l. 95: this fact deserves a reference

Response: We added two references that deal with spatio-temporal nowcasting of precipitation (Shi et al. 2015, Tian et al. 2019).

p.5, l. 116: this reference is given twice, please remove one

Response: We corrected this.

p.9, l.189 – p.10, l. 191: please specify the hardware resources you used and give advice on some minimal requirement

Response: We added a "Technical Implementation" subsection in the methodology section where we specify the hardware resources and a framework description. It is, however, difficult to give some minimal requirements. In theory, this model could be run on any laptop after training.

p.10, l. 193: this may be the true final step, while "final" was already used on p. 7, l. 174. Please correct this.

Response: We changed this to "In another experiment …".

Figure 1: Please add to the caption the data source and reference to the product.

Response: We added a reference to DWD RADKLIM here.

Figure 7. The caption is the same as the one of Figure E1 although the maps are different. Please check the captions.

Response: We adjusted the caption of Figure E1, to indicate that this is the reconstruction for the entire area of Germany; and Fig. 7 is only a zoomed in portion. Additionally, in the caption of Fig. 7, we indicated that Fig. E1 includes the entire territory of Germany.

Anonymous Referee #2

**General comments**

The authors propose and analyze a new approach to the interpolation of values for incomplete precipitation maps. The authors motivate their study by explaining that such interpolation is often necessary as result of various factors, such as radar outages. They explain the structure of their "channel-based" approach, combined with an LSTM module, and apply it to a dataset provided by Germany's national meteorological service. Finally, they show via a range of metrics and modeling scenarios that their proposed approach may outperform more basic ones.

The overall quality of the writing, referencing, structure and experiments is good. There do not appear to be any notable and relevant works missing from the list of references. Although not providing ground-breaking contributions to the field, the results are nonetheless valuable in illustrating the feasibility of alternative approaches to the problem of precipitation map in-painting using temporal data. The description and review of the addressed problem is also good. The manuscript would be suitable for publication in the journal after minor revisions to address the following concerns and questions. Particularly notable insightful and clear parts of the manuscript are also noted below.

Response: we thank the reviewer for the review and for these encouraging remarks. Regarding novelty, we do believe that this approach is rather new, as the ML method allows us to 1) take account of moving weather systems in a dynamical fashion, 2) were able to update the entire existing catalogue of the German weather service, and 3) provide a potential method to quickly infill rainfall information when radar systems fail, for instance in emergency situations. We will add text to the paper, to stress these points.

**Review ratings**

Scientific significance: Good

Scientific quality: Good

Presentation quality: Fair

**Specific and technical comments**

Line 24: Unclear if this is in fact referring to Figure 1.

Response: indeed, we meant to refer to Fig. 1, we adjusted this in the revised paper.

Line 40: More detail should be provided to explain what is meant by "data-driven". I assume such methods are machine learning adjacent ones which contrast to statistical methods driven by modeling assumptions, but additional clarity would be helpful here.

Response: we added the difference netween numerical models and/or statistical and geoinformation techniques based on fixed equations, rather than learning algorithms based on large datasets where the equations are learned by training (which is "data-driven").

Line 47: Additional detail would be beneficial here. It appears unfair to state that data-driven methods only account for spatial relationships while statistical methods only account for temporal variability. Explanation of the caveats of this statement should be included. For example, why is spatial interpolation, a statistical method, unable to reconstruct spatial features?

Response: we meant to indicate that temporal changes in parts of the spatial data (images) are used to learn about infilling in missing parts. This is the case with "moving" weather systems that drive rainfall patterns. This is a feature that is not included in geostatistical methods. We adjusted this sentence to make that clear.

Line 59: The explanation of the dataset is clear and well organized. However, it is unclear which exact time period is used for this study.

Response: we added the years for which we used and updated the data, which is the period 2001-2022.

Line 85: Please provide a citation for this statement.

Response: We added Liu et al. 2018, where they compare their partial convolutions with other techniques and show an improvement of their technique.

Equations 1: Please use \mathrm to format the text

Response: We corrected this.

Line 102: This paper seems to be a seminal work in the field, and the overall approach of the presented work appears rather similar. Since many readers would already have this particular paper in mind when reading through this manuscript, it would be ideal to mention this paper earlier in the introduction and explain its similarities and differences with this work.

Response: Thanks for this suggestion, and we agree. We added a sentence in the introduction, where we explain that our method is a combination of the work from Liu et al. 2018 and Shi et al. 2015, combining partial convolutions and convolutional LSTMs.

Line 122: There should be a brief mention of the programming language and packages used to implement the model, hardware utilized, and run time here.

Response: we thank the reviewer for this comment. We added a "Technical Implementation" subsection in the methodology section, where we go into detail here. We added training and inference times in the discussion section.

Line 125: The experiments and associated results in this section are highly yet appropriately detailed. However, it is difficult at times to follow what is going on. Some reorganization here could be beneficial to the reader's understanding. For example, the results and discussion could be split into at least two different sections.

Response: We added a subsection "Experimental Design" to the methodology section and moved all parts that describe the experiments to this. We hope this is beneficial to the reader's understanding.

Line 137: Please use \mathrm to format the text

Response: We corrected this.

Figure 2: This is a good way to present the relevant metrics in a visually appealing and informative manner

Response: we thank the reviewer for this compliment.

Figure 7: The text in this figure is difficult to read as it is too small.

Response: we will check the font size and also ask the journal about requirements.

Line 198: The conclusion is clear and provides a high-level summary of the main findings. A very brief reiteration of the motivation for the study here would also be helpful.

Response: we added a reiteration for the scope and motivation of the study in the revised paper.